# W-TSS: A Wavelet-Based Algorithm for Discovering Time Series Shapelets

**DOI:** 10.3390/s21175801

**Published:** 2021-08-28

**Authors:** Kenan Li, Huiyu Deng, John Morrison, Rima Habre, Meredith Franklin, Yao-Yi Chiang, Katherine Sward, Frank D. Gilliland, José Luis Ambite, Sandrah P. Eckel

**Affiliations:** 1Department of Population and Public Health Sciences, University of Southern California, Los Angeles, CA 90032, USA; Jmorr@usc.edu (J.M.); habre@usc.edu (R.H.); meredith.franklin@usc.edu (M.F.); gillilan@usc.edu (F.D.G.); eckel@usc.edu (S.P.E.); 2Spatial Sciences Institute, University of Southern California, Los Angeles, CA 90089, USA; 3Applied AI and Data Science, City of Hope National Medical Center, Duarte, CA 91010, USA; huiyuden@usc.edu; 4Department of Computer Science and Engineering, University of Minnesota, Minneapolis, MN 55455, USA; yaoyic@usc.edu; 5Department of Biomedical Informatics, University of Utah, Salt Lake City, UT 84108, USA; kathy.sward@nurs.utah.edu; 6Department of Computer Science, University of Southern California, Los Angeles, CA 90089, USA; ambite@isi.edu

**Keywords:** shapelets, wavelets, time series mining, time series classification, pattern discovery

## Abstract

Many approaches to time series classification rely on machine learning methods. However, there is growing interest in going beyond black box prediction models to understand discriminatory features of the time series and their associations with outcomes. One promising method is time-series shapelets (TSS), which identifies maximally discriminative subsequences of time series. For example, in environmental health applications TSS could be used to identify short-term patterns in exposure time series (shapelets) associated with adverse health outcomes. Identification of candidate shapelets in TSS is computationally intensive. The original TSS algorithm used exhaustive search. Subsequent algorithms introduced efficiencies by trimming/aggregating the set of candidates or training candidates from initialized values, but these approaches have limitations. In this paper, we introduce Wavelet-TSS (W-TSS) a novel intelligent method for identifying candidate shapelets in TSS using wavelet transformation discovery. We tested W-TSS on two datasets: (1) a synthetic example used in previous TSS studies and (2) a panel study relating exposures from residential air pollution sensors to symptoms in participants with asthma. Compared to previous TSS algorithms, W-TSS was more computationally efficient, more accurate, and was able to discover more discriminative shapelets. W-TSS does not require pre-specification of shapelet length.

## 1. Introduction

Time series classification methodology is of growing interest in health research, especially given recent advances in sensor technology. For example, environmental health researchers may be interested in using daily exposure time series to distinguish between days a study participant does or does not report respiratory symptoms. Many time series classification methods distinguish between classes using global summary statistics (e.g., mean, standard deviation) or global shapes (e.g., dynamic time warping methods) [1,2]. However, there may be discriminative local shapes (e.g., peaks in exposure indicating proximity to a source) missed by methods using global summaries. One promising method using local features is time series shapelets (TSS), first introduced by Ye and Keogh [3]. Shapelets are defined as maximally discriminative subsequences of a set of labelled time series. TSS classifies time series based on similarity to local shapes and has the potential to outperform other state-of-the-art time series classifiers using global features, especially in applications with discriminative local shapes and in the presence of general noise and distortion [4].

Shapelets have only recently been applied to studies of human health, with applications including identification of: temporal patterns (over months) of regional PM^2.5^ and PM^10^ related to US counties with higher or lower lung cancer incidence [5], temporal patterns (over hours) of heart rate, respiratory rate, and systolic blood pressure predictive of the severity of future sepsis events in ICU patients [6], and temporal patterns (over hours) in sequential organ failure assessment score related to mortality in ICU patients [7]. The discriminative local shapes identified by TSS have the potential to be of scientific interest (e.g., peaks related to exposure to air pollution sources).

TSS algorithms can be summarized by the following basic steps: (1) identify a candidate shapelet S defined as a contiguous time series subsequence of length *L* starting at position *i*, which can be written as $S\;={\;t}_{i},t_{i+1},\;\ldots ,{\;t}_{i+L-1}$, (2) calculate the Euclidean or dynamic time warping distance between *S* and all possible subsequences of the same length from each time series in the training data, (3) calculate the minimum distance between *S* and all subsequences of the same length from a given time series T, (4) repeat steps 1-3 for many candidate shapelets, (5) use the minimum distances to the large set of candidate shapelets as features and build a machine learning model (e.g., tree-based classifier) to discover the most important features (i.e., shapelets) for predicting the class of each time series.

Identifying shapelets is the most computationally intensive aspect of TSS algorithms due to the huge number of potential candidate shapelets of a given length *L*, and the even larger number when *L* is tuned. Using exhaustive search to discover shapelets with a given length *L* requires examination of (*ATL* + 1 − *L*)*n* candidate shapelets, where *ATL* denotes the average total length of all time series and *n* is the number of time series. For example, in a dataset with 1000 time series each of length 1440 (number of minutes in a day), there would be 1.381 million potential candidate shapelets of length 60 and 1.421 million potential candidate shapelets of length 20. Previous efforts to speed up shapelet discovery can be summarized into three categories: (1) upscaling the time series by aggregating some continuous timestamps of the time series to reduce the average total length (e.g., using symbolic aggregate approximation (SAX) to convert the time series and then using the SAX conversions to calculate the similarities and to find the shapelets [8]), (2) using a certain threshold to prune or sample from the shapelet candidates (e.g., [9] proposed a fast shapelet discovery algorithm based on important data points (IDPs) and only the subsequence containing one or more identified IDPs will be selected as a candidate shapelet), (3) building neural networks or other learning approaches to learn the shapelets with learning objectives of minimizing the distances between the time series and the candidate shapelets [10,11,12,13]. In the third method, the final shapelets are no longer actual subsequences of observed time series in the training data, but rather optimized sequences based on either actual subsequences or cluster centers of actual subsequences (e.g., k-means cluster centers).

These fast shapelet discovery algorithms each have limitations. For example, upscaling/aggregating methods may lose temporal details due to the coarser temporal granularities, pruning/sampling methods have larger chance of missing the real underlying maximally discriminative shapelets, and learning methods can be sensitive to the initial values of the candidate shapelets. Additionally, all these methods rely on the proper tuning of the hyperparameter *L* (shapelet length). To discover the most discriminative subsequences of the time series, various lengths of shapelets need to be tested. In this study, we propose a novel approach (W-TSS) that leverages wavelet transformations to intelligently and quickly discover shapelets of various lengths. We tested W-TSS in three case studies with increasingly difficult classification tasks: (1) discriminating four classes of synthetic time series from the UCR Time Series Archive [14], (2) discriminating indoor vs. outdoor PM^2.5^ time series from measurements made at the residences of participants in a pediatric asthma panel study conducted by our research group, and (3) in the same panel study discriminating days with vs. without asthma inhaler usage based on residential indoor PM^2.5^ time series. In Section 2, we describe the proposed automated method combining wavelet transformations, extraction of candidate shapelets, and machine learning predictions. In Section 3, we compare W-TSS with learning TSS in Task 1 and examine the performance of W-TSS in Tasks 2 and 3. In Section 4, we discuss the strengths and limitations of W-TSS.

## 2. Materials and Methods

### 2.1. Learning TSS

We will use learning TSS as a comparator to W-TSS, so we briefly summarize the learning TSS algorithm proposed by Grabocka et al. [13]. Learning TSS focuses on learning a local discriminative pattern rather than identifying it through search. The algorithm begins with an initial guess of the shapelets, which could either be arbitrary values or results from preliminary data exploration (e.g., k-means cluster centers from a sample of subsequences of the pre-specified length). The number of shapelets to be initialized and their length are set as hyperparameters. At each step of the algorithm, we calculate a Soft-Minimum Distance matrix (*M*) between each of the current shapelets and each of the observed time series. The distance between a shapelet and a time series is defined as the minimum distance between the shapelet and all sliding window segments of that size from that time series. Then *M* is used to predict the time series classes in a regularized linear model. A stochastic gradient descent algorithm is used to calibrate (update) the shapelets as weights, and the algorithm repeats until convergence. Figure 1 demonstrates the iterative process of learning TSS. The upper left panel shows an observed time series (from the TRACE data, described later) in blue, and the best matching locations of eight shapelets (multiple colors), four with fixed length of 25 and four with fixed length of 50. The upper right panel zooms into a subsequence of the original time series, and the lower panels demonstrate, for three selected shapelets, the step-by-step progression from initialized values (blue) to final values (red) in the learning TSS algorithm. Note that all three of these shapelets contain a sharp decreasing shape which matches with the decrease in the observed time series at time ~217.

### 2.2. Wavelet-Based Discovery for TSS (W-TSS)

The wavelet transform is a signal processing method developed as a localized alternative to the Fourier transform [15]. The Fourier transform identifies global frequencies using sums of infinite sinusoidal functions and can perform poorly in time series with certain characteristics, including discontinuities and sharp spikes. In contrast, the wavelet transform identifies local frequencies present in a time series and the time at which these frequencies occurred. The wavelet transform has proven very useful in analyses of time series with brief characteristic oscillation (e.g., electrocardiography (ECG)) [16]. Wavelets are functions constructed with specific mathematical properties which begin and end at zero, with a brief wave-like oscillation in between. For example, the Morlet wavelet as a member of continuous wavelets mathematically is composed of a complex exponential function multiplied by a Gaussian window. In comparison, Daubechies wavelet as one of the most used discrete wavelets is based on the use of recurrence relations to generate progressively finer discrete samplings from an implicit mother wavelet function. Examples of some iconic wavelet functions are shown in Figure 2.

A wavelet transformation can be written in the following general form:(1)$$Y\left(a,\;b\right)=\frac{1}{a^{1/2}}\int_{-\infty }^{\infty }X\left(t\right)¦\times \left(\frac{t-b}{a}\right)dt$$
where *¦×* is the wavelet function which gets scaled by factor *a* and translated by factor *b*, *Y* is the transformed time series, and *X* is the original time series. The scale factor *a* is inversely proportional to the frequency of the wavelet function and has a similar meaning to period, commonly used in other time series analysis. Conversion to pseudo-frequencies is possible by taking: $f_{a}=f_{c}/a$, where $\;f_{a}$ is the pseudo-frequency, and $f_{c}$ is the central frequency of the wavelet function. Wavelet function families are divided into continuous (as examples shown in the bottom row of Figure 2) and discrete (as examples shown in the top row of Figure 2) based on whether the scale can be decimal numbers. For discrete wavelet transformation, the scale factor *a* increases by powers of two (e.g., *a* = 1, 2, 4,...) and the translation factor *b* increases by integer values (e.g., *b* = 1, 2, 3,...). Since all wavelet functions must be finite in energy, there always exists a “window” of non-zero magnitude in the wavelet function. The translation factor *b* is how far we slide the “window” from the starting of the time series. Different wavelet functions have their own unique suitability to examine different types of abrupt changes that occur in the signals.

The wavelet transforms outputs both frequency-domain and time-domain information. This is accomplished by working with a set of different scales (*a*) from large to small. A scalogram is a representation of the 2-dimensional time-scale output of the wavelet transformation of a 1-dimensional signal. Figure 3 shows scalograms for four different wavelet transformations (top four panels) applied to a single time series from the TRACE dataset (bottom panel). Note that the time series is generally flat with a single large increase centered at time ~100, with a total duration of ~20. If we pick a scale and segment the time series according to its scalogram power (higher power indicated by red in Figure 3), we can find subsequences where the time series has high energy of that scale. For example, all four wavelet functions in Figure 3 identify high power for low scale signals (~32) occurring near time 100, though the Mexican Hat and Morlet wavelet function identify the low scale on either side of 100 while Gaussian and Complex Gaussian identify the low scale signal at ~100. By picking a scale of 32 in Figure 3, we find subsequences corresponding to where the time series increased at time ~100. For the Morlet and Mexican Hat wavelet transformations, we would have ~2 subsequences clipped near 100 for candidate shapelets, whereas for the Gaussian and Complex Gaussian transformations, we would have a single subsequence. This result highlights the importance of trying different wavelet function.

This intuition motivates our approach for W-TSS. To automate the process of extracting subsequences with high energy at the identified scales for W-TSS, we convert the scalogram to a binary image containing only 0′s and 1′s according to the following procedure: (1) for a given scale, standardize the power values of the transformed time series, (2) set a threshold hyperparameter (1 as default, but could be modified), and (3) dichotomize the scalogram with 0 indicating the standardized value < threshold and 1 indicating ≥ threshold. Based on the dichotomized scalogram, we extract the subsequences of 1s, which represent the parts of the time series having higher energy at the selected scales. To minimize the inclusion of short subsequences arising from noise, we also set a minimum length threshold to remove subsequences with insufficient length. This is especially important with small scales which have the potential to introduce many candidates from random noise. With the extracted subsequences from the binary scalogram, we created the initial pool of W-TSS candidate shapelets. Based on the size of the initial pool, we could either directly use all candidate shapelets or reduce the number of candidate shapelets (Figure 4 shows the workflow chart of selecting shapelets using W-TSS).

In the following data analysis sections, we implemented W-TSS using: complex Gaussian wavelet functions, two selected scales (based on visualization of scalogram), minimum length of 3 to filter out noises; and reduced the number of resultant candidate shapelets using two simple methods: a variant of k-means clustering or by filtering out low variation shapelets. Similar to other TSS-based algorithms, we finally built machine learning models using input features based on Euclidean distances (future work could use DTW) between the final TSS and the input time series. For the simple synthetic data example in Section 3.1 we used a multiclass logistic regression model. For the more complex real data analyses in Section 3.2.1 and Section 3.2.2, we used a gradient boosted trees classifier (Xgboost) [17]. Key hyperparameters and their tuning ranges are listed in Table A1.

### 2.3. Datasets

Synthetic TRACE dataset from UCR Time Series Archive. The TRACE dataset contains four classes of synthetic time series designed to simulate instrumentation failures in a nuclear power plant [18]. Each class of time series has unique local discriminative patterns, but with shifts in the exact timing of the local patterns for each time series in a given class (Figure 4). The length of each time series is 275. There are 100 time series in the training dataset and 100 time series in the test dataset, roughly balanced by class.

Pediatric Research Using Integrated Sensor Monitoring Systems (PRISMS) dataset from the Utah Informatics Platform Center. A panel study of 10 participants with asthma (ages 5-51, four children and six adults) in seven households near Salt Lake City, Utah was conducted from April 2017 to April 2018. The study was approved by the University of Utah Institutional Review Board (IRB) with study number IRB_00086107. The original approval dated 28 April 2016, and the study maintained ongoing approval with the most recent approval on 21 March 2020. Data were shared as per the PRISMS Consortium Data Use Agreement with the University of Southern California and other PRISMS collaborators. Participants signed informed consent documents that included sharing the data used in this analysis. Particulate matter air pollution less than 2.5 microns in aerodynamic diameter (PM_2.5_) was measured both inside and outside of each household. Both indoor and outdoor PM_2.5_ were collected by the deployed sensors (commercial Dylos Corporation particle counters, modified to include sensors for humidity and temperature, and to include wi-fi communications). The conversion of particle counts to mass concentrations (in μg/m^3^) followed rules suggested previously [19]. We then separated the sensor signals into daily time series, with each day starting at 8 p.m. (the approximate time of the patients to take the questionnaires) and ending at 8 p.m. the next day. Participants (or their guardians, for some child participants) from these households were asked to submit daily questionnaires about their asthma symptoms and medication usage in the past 24 h, including frequency of use of rescue medication (“How often did you use an albuterol or Xopenex inhaler or received a nebulized treatment in the last 24 h?”). There were 823 days with complete data for indoor and outdoor minute-level PM_2.5_ exposure time series and asthma medication use. In Section 3.2.1, we discriminated between indoor vs. outdoor PM_2.5_ time series (each day treated as a separate 24 h-long time series). In Section 3.2.2, we discriminated between days with and without rescue medication use based on the corresponding 24-h residential indoor PM_2.5_ time series before the daily submitted questionnaires. For each PRISMS data application, we randomly selected 33% of days for holdout test data (311 testing data samples in Section 3.2.1 and 272 testing data samples in Section 3.2.2). We used random five-fold cross validation to train the model and evaluated performance of the final model in the test data.

## 3. Results

### 3.1. Synthetic TRACE Dataset

Applying W-TSS with scales of 32 and 64 to the TRACE dataset produced candidate shapelets of various lengths (Figure 5, right panel). There were clearly different groups of candidate shapelets, and each group appeared to represent an iconic local shape from the four time series classes.

While it could be feasible to input all candidate shapelets into a prediction model which can handle high dimensional data, the clear groupings of the candidate shapelets suggested that preliminary dimension reduction would be reasonable and would likely improve interpretation. We reduced the number of candidate shapelets using global alignment kernel k-means [20], with the number of clusters arbitrarily set to k = 12. Our final set of candidate shapelets were the center lines of the 12 resultant clusters (Figure 6).

The minimum distances of each time series to each of the 12 shapelets were used as features in a multiclass logistic regression model fit to the training dataset. In the test dataset, time series from all four classes were perfectly classified (Table 1).

The synthetic TRACE data is well-suited to TSS applications, and both W-TSS and learning TSS were highly accurate. Grabocka et al. have previously applied learning TSS to the TRACE data and reported a test accuracy of 98% [13]. However, the shapelets identified by W-TSS better matched the iconic local patterns of the four time series classes than did the shapelets identified by learning TSS (Figure 7). Each of the W-TSS shapelets closely matched a subset of classes of time series and poorly matched other classes, indicating better discriminative ability, whereas the shapelets by learning TSS are more randomly matched to places of the original timeseries

### 3.2. PRISMS Dataset from Pediatric Asthma Study

Unlike the synthetic TRACE data, the real-world PRISMS PM_2.5_ data show much more variation in both global and local patterns, posing a greater challenge for the classification tasks.

#### 3.2.1. PRISMS: Daily Indoor PM_2.5_ vs. Outdoor PM_2.5_ Time Series

The set of daily outdoor PM_2.5_ time series had higher variation and fewer apparent baselines than the set of indoor PM_2.5_ time series (Figure 8).

After applying W-TSS with scales of 256 and 512, we extracted a total of 6179 candidate shapelets: 2983 from indoor PM_2.5_ time series and 3196 from outdoor PM_2.5_ time series (Figure 9, top panels). Alternatively, a brute force method using two fixed lengths of 256 and 512 would have produced 975,255 candidate shapelets of length 256 for both indoor and outdoor PM_2.5_ and 764,567 candidate shapelets of length 512 for both indoor and outdoor PM_2.5_. W-TSS rapidly identified <0.2% of those total possible candidate shapelets, dramatically reducing computational load.

However, 6,179 candidate shapelets is still a large number of input features, especially given that the training data has much less (1646) time series. Reducing the number of candidate shapelets using clustering methods (e.g., global alignment kernel k-means) would not be appropriate here given the large variation in shapes. However, we noticed that many of 6,179 candidate shapelets were quite “flat”, with similar levels in both classes. Hence, we decided to filter by removing low variation candidate shapelets. We arbitrarily chose a threshold on the candidate shapelet variance which retained 20 candidate shapelets from the outdoor PM_2.5_ time series. Applying this same threshold to the indoor data retained 59 candidate shapelets from the indoor PM_2.5_ time series (Figure 9, bottom panels). Filtering by variance may have eliminated some truly discriminatory shapelets, but in this application we were more interested in high-variation shapelets. Future analyses might consider combining the unsupervised W-TSS candidate shapelet discovery with other supervised TSS discovery methods. An Xgboost model, using minimum distances to the 79 shapelets (59 indoor, 20 outdoor) as input features, perfectly classified the test data (Table 2).

Feature importance of the top-10 shapelets is shown in Figure 10 (Figure A1 in Appendix A shows feature importance of all the shapelets). Shapelets were assigned numerical names according to the ranking of the shapelets’ variance from 0 to 78, with a low number indicating larger variance. The most important shapelets were neither the most nor the least variable (e.g., variance ranking of the top six shapelets was 11 to 48). The top six shapelets (Figure 11, top panels) represented transient spikes in PM_2.5_ (Shapelets 11 and 13), a transient spike followed by a gradual decline (Shapelet 48) and sharp increases (Shapelets 30, 35, 31). Partial dependence plots (Figure 11, bottom panels) indicated that PM_2.5_ time series with close matches to each of these top six shapelets had a higher probability of being from the indoors (vs the outdoors). We inspected the top six shapelets and confirmed that they had all been identified from indoor PM_2.5_ time series. Given these results (top shapelets coming from indoors time series and perfect classification in test data), we conclude that in the PRISMS data indoor PM_2.5_ time series tend to display unique discriminative local patterns not observed in outdoor PM_2.5_ time series.

#### 3.2.2. PRISMS: Daily Indoor PM_2.5_ Time Series with and without Rescue Medication Usage

Rescue medication use was reported on 60 (7.3%) of the 823 days. There were no obvious differences between residential indoor PM_2.5_ time series with and without rescue medication usage (Figure 12).

After applying W-TSS with scales of 256 and 512, we extracted 2815 candidate shapelets from indoor PM_2.5_ time series on the 763 days without rescue medicine use, and 168 candidate shapelets from the 60 days with use (Figure 13, top panels). The imbalance in the number candidate shapelets was due to the imbalance in rescue medicine use days. In addition, note that the 2815 plus 168 candidate shapelets are the same as the 2983 candidates for indoor PM_2.5_ from Section 3.2.1 since the settings of our unsupervised W-TSS discovery algorithm were the same. To reduce the size of the set of candidate shapelets, we again applied a variance filter selecting a threshold to retain at least 20 candidate shapelets with high variation in the minority class (days with rescue medicine use). Applying the same filter to days without rescue medicine use retained 38 candidate shapelets (Figure 13, bottom panels). Note that our simple approach to determining a filtering threshold is supervised and could be substituted by more complex supervised methods.

The Xgboost model using minimum distance to the 58 shapelets as input features had relatively poor performance in the test dataset, with an F_1_ score of 0.26 and only four out of 12 rescue medicine use days correctly classified (Table 3). Figure 14 shows the feature importance of the top-10 shapelets (Figure A2 shows feature importance of all the shapelets). The top six shapelets all represented patterns of quick to more gradual increases in PM_2.5_ concentrations of at least an hour in duration (Figure 15, top panels). Partial dependence plots (Figure 15, bottom panels) only showed a clearly increased probability of a rescue medicine use on days with close matches to Shapelet 42, which was the shapelet with the highest feature importance. Additionally, the minimum distances of the time series from the shapelets were small (<0.02) compared to what was observed in Section 3.2.2 (between 0 and 10), suggesting that all the time series have a relatively close match to the top shapelets. This likely indicates the lack of a single discriminative local pattern for rescue medication use, and partially explains the poor prediction performance.

## 4. Discussion

In this study, we demonstrated in the synthetic TRACE dataset that W-TSS produced a reasonably sized set of candidate shapelets (<0.2% of the brute force method) and led to a set of final shapelets which more closely matched the discriminative iconic local patterns and had better (perfect) accuracy than learning TSS. In the PRISMS panel study data, W-TSS was able to perfectly predict indoor vs. outdoor PM_2.5_ daily time series. In the more challenging application of identifying days with rescue medication use using daily residential indoor PM_2.5_ time series, W-TSS identified patterns of exposure (even though not discriminating), which could be investigated in future studies. The association between asthma rescue medicine use and residential indoor PM_2.5_ is complex and careful study of this association requires accounting for potential confounding variables and exposure misclassification, including time spent away from home. It is not expected that indoor PM_2.5_ can perfectly predict asthma medication use, but the patterns of PM_2.5_ discovered by the data-driven W-TSS method are suggestive of indoor sources and are of scientific interest.

In many applications with small number of data samples, it may make sense to filter the initial set of candidate shapelets produced by W-TSS before they are input as features in machine learning algorithm for predicting the time series classes. We implemented two such methods: a variant of k-means clustering (global alignment kernel k-means) to find common representative shapelets and a simple filter to remove shapelets with low variance. When implementing the k-means clustering, the number k of clusters must be specified in advance. Thus, there would be risks of the chosen number k not reflecting the data. Moreover, as a nonparametric algorithm, the outcomes depend on the initial cluster centers. To over the challenges, a useful heuristic is to use several random initial center assignments and select the best result according to some criteria, e.g., using the intraclass inertia or using hierarchical clustering algorithm in conjunction with k-means [21]. Other reduction methods could also be applied (e.g., removing highly correlated shapelets or all the initial W-TSS candidate shapelets could be input into a machine learning algorithm tailored for supervised feature selection). The W-TSS approach for candidate shapelet discovery could also be combined with features of other TSS methods. For example, W-TSS candidate shapelets could be pro-vided as initial values in the learning TSS algorithm.

Besides of wavelet transformation, there are other popular time-frequency decomposition methods for processing time series, e.g., Short-Time Fourier Transform, Hilbert–Huang Transform, Constrained Least-Squares Spectral Analysis, and Least-Squares Wavelet Analysis [22]. Future work will focus on evaluating different time-frequency decomposition methods, providing more accurate and reliable estimates of change or breakpoint detection especially in non-stationary time series, as well as more intelligent methods to choose scales based on the time-frequency decomposition (e.g., the scalogram). Future work will also involve extraction of multivariate patterns for the needs of using complex high-dimensional temporal data to support health care decisions. A promising future investigation will combine W-TSS with existing multivariate shapelets discovery algorithms based on fast shapelets discovery algorithms [23] to more efficiently identify multivariate shapelets.

## 5. Conclusions

This study focused on extraction of univariate interpretable patterns from time series and developed the novel W-TSS approach for unsupervised discovery of candidate shapelets for TSS using wavelet transforms with key acronyms listed in Table A2. Discovering the localized temporal patterns could be extremely important in many realms (i.e., environmental health) that needs to associate timeseries data with sparse outcomes. Compared to the other methods, the advantages of W-TSS include: (1) no loss of time resolution compared to the other TSS algorithms using aggregation/upscaling, (2) the initial candidate shapelet discovery is unsupervised leading to greater computational efficiency since there is no need to run a machine learning model, and (3) no need to pre-specify the length of the candidate shapelets. Even though the wavelet function and scales do need to be specified, but these can be informed through examination of the scalogram. The examination of several wavelet functions and several different randomly selected scalograms is recommended, but this procedure is still less time consuming than tuning the fixed lengths of the candidate shapelets. In summary, W-TSS offers a computationally efficient unsupervised method for automatic discovery of candidate shapelets of different lengths in TSS without degrading temporal resolution.

## Figures and Tables

**Figure 1 sensors-21-05801-f001:**
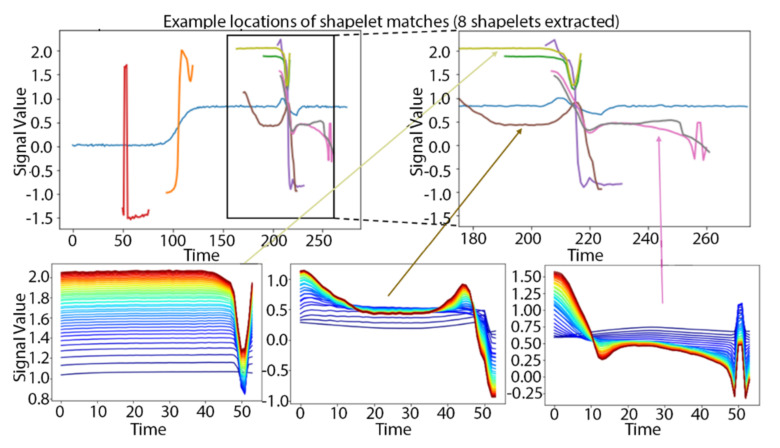
Example of applying Grabocka’s learning TSS method to identify 8 shapelets in the TRACE data. The bottom three subplots demonstrate the step-by-step learning of three selected shapelets, with blue denoting the initial value and red denoting the final trained value of the shapelets.

**Figure 2 sensors-21-05801-f002:**
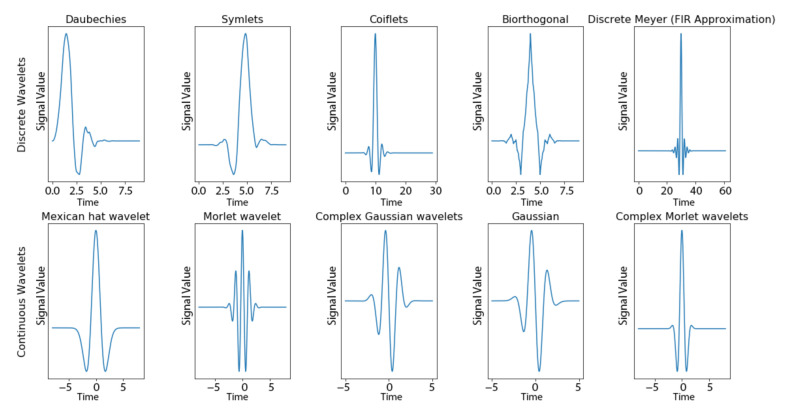
Examples of wavelet functions that are discrete (**top row**) or continuous (**bottom row**).

**Figure 3 sensors-21-05801-f003:**
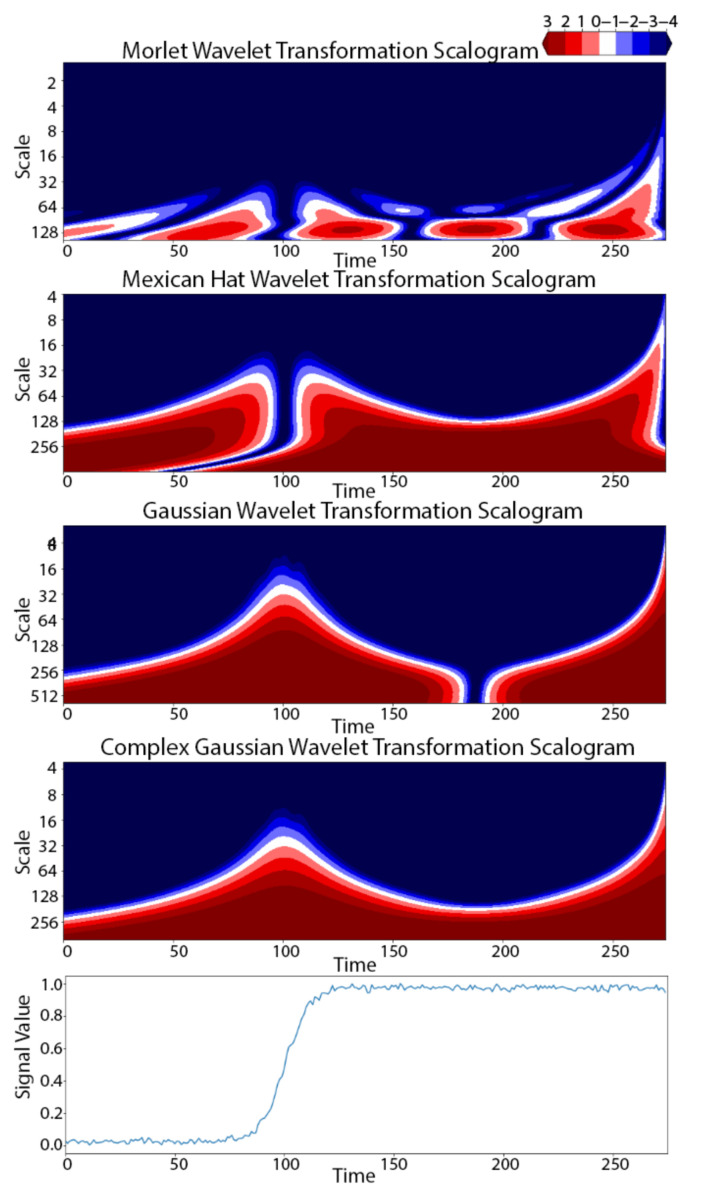
Scalogram representation of the 2-dimensional time-scale output of four different wavelet transformations (**top four panels**) of a single time series (**bottom panel**) from the TRACE dataset. The red color shows where the power of the transformed signal is concentrated.

**Figure 4 sensors-21-05801-f004:**
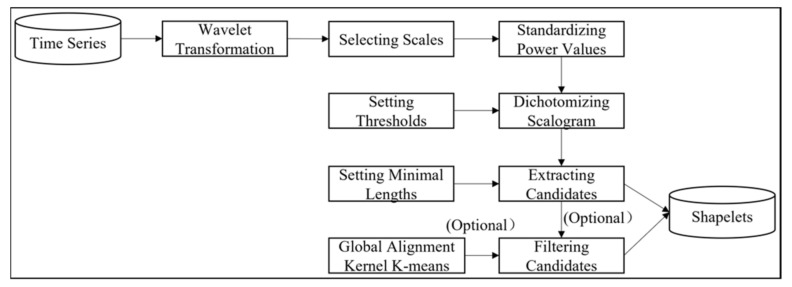
Workflow chart of applying W-TSS to select shapelets.

**Figure 5 sensors-21-05801-f005:**
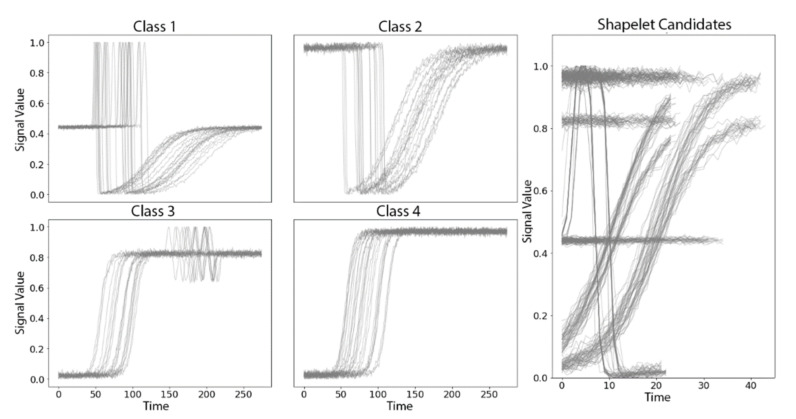
Time series from the four classes in the synthetic TRACE dataset (**left panels**) and the initial set of candidate shapelets identified by W-TSS (**right panel**). X-axis: time, Y-axis: risk of instrumentation failures.

**Figure 6 sensors-21-05801-f006:**
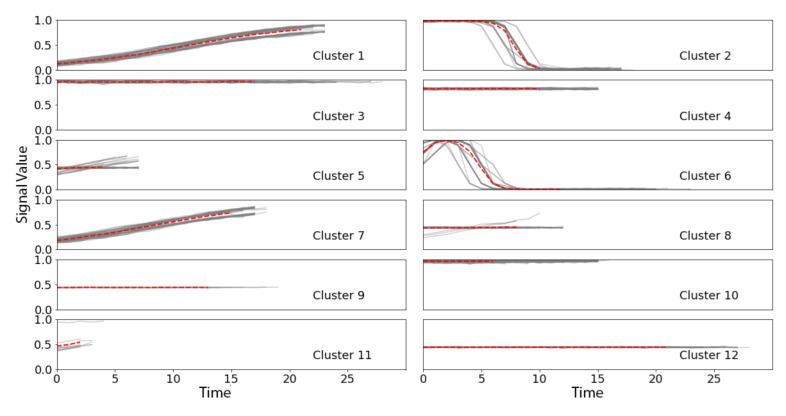
Final set of 12 shapelets for W-TSS in the TRACE dataset, identified as the center lines (red) from global alignment kernel k-means clustering results on the W-TSS candidate shapelets, with time series members (grey solid line) and centerlines (red dash line). X-axis: time, Y-axis: risk of instrumentation failures.

**Figure 7 sensors-21-05801-f007:**
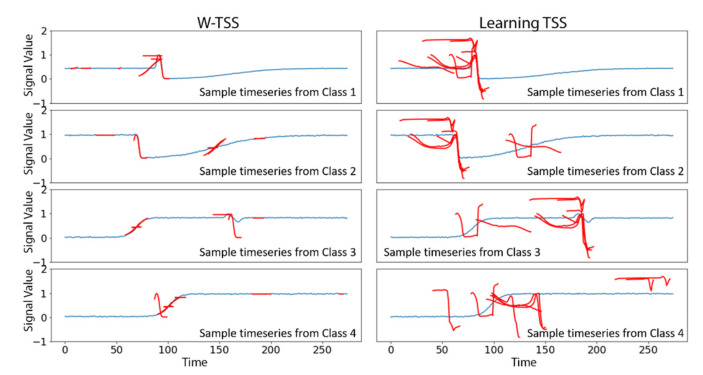
Shapelets (in red) discovered in the TRACE dataset by W-TSS (**left**) and learning TSS (**right**), displayed at the best matching location of an example time series (in blue) from each of the four classes.

**Figure 8 sensors-21-05801-f008:**
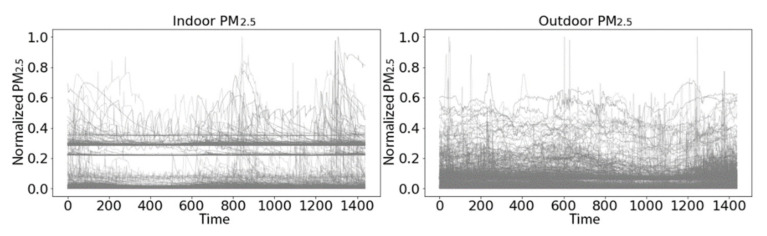
The 823 daily indoor (**left**) and outdoor (**right**) PM_2.5_ time series, each 1440 min long (24 × 60), with the PM values scaled to min-max to plot on a unit interval.

**Figure 9 sensors-21-05801-f009:**
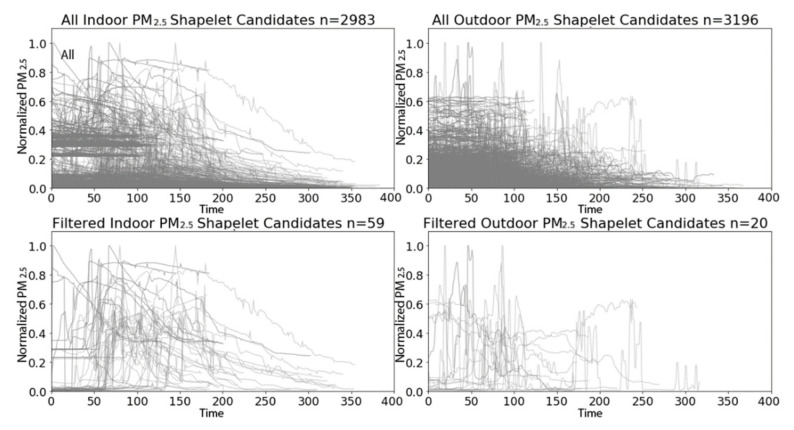
W-TSS shapelets for the PRISMS indoor and outdoor PM2.5 time series, before filtering (**top row**) and after filtering (**bottom row**).

**Figure 10 sensors-21-05801-f010:**
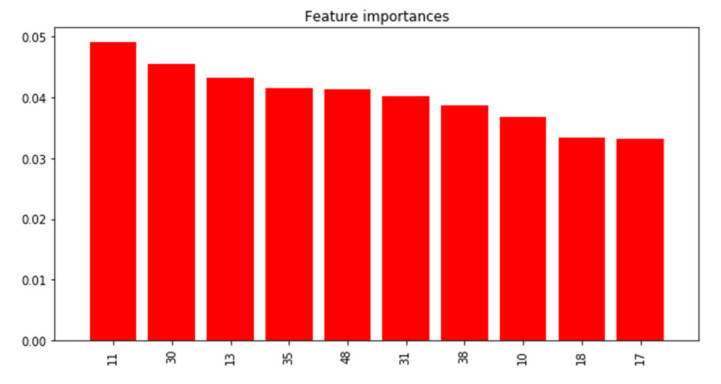
Feature importance for top-10 W-TSS shapelets in predicting indoor vs. outdoor time series PM_2.5_ in PRISMS.

**Figure 11 sensors-21-05801-f011:**
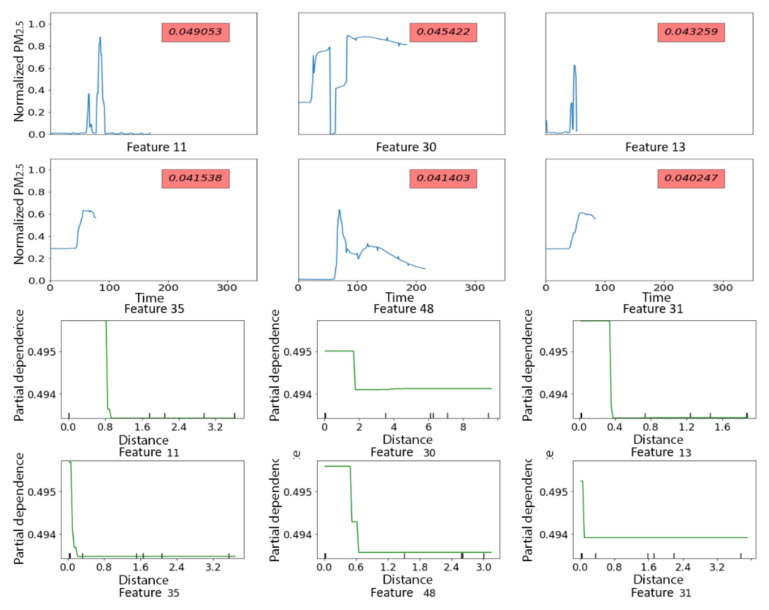
Top 6 shapelets predicting indoor vs. outdoor PM_2.5_ in PRISMS. **Top** panels display the shapelets along with their feature importance and **bottom** panels display partial dependence plots with x-axes representing the minimum distance to the shapelet and y-axies representing predicted probability of indoor (vs. outdoor).

**Figure 12 sensors-21-05801-f012:**
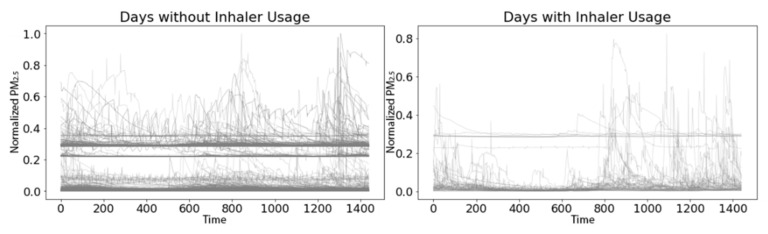
Daily indoor PM_2.5_ time series from the 763 days without rescue medicine use (**left**) and the 60 days with rescue medicine use (**right**). Each time series is 1440 min long and the PM values have been min-max scaled to lie on the unit interval.

**Figure 13 sensors-21-05801-f013:**
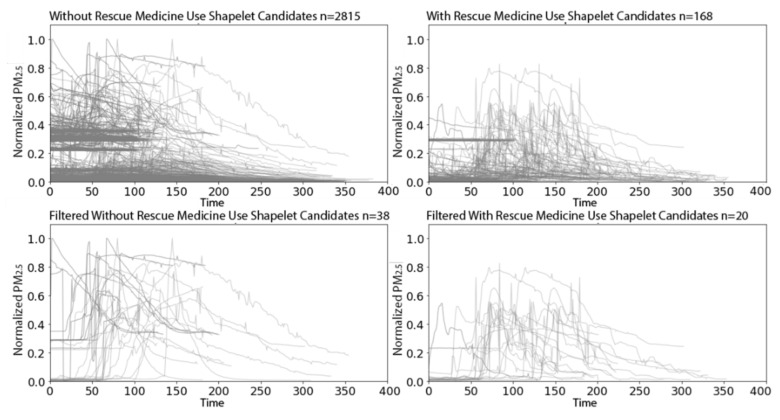
W-TSS shapelets from indoor PM2.5 time series before filtering (**top row**) and after filtering (**bottom row**) for days without rescue medicine use (**left**) and days with use (**right**).

**Figure 14 sensors-21-05801-f014:**
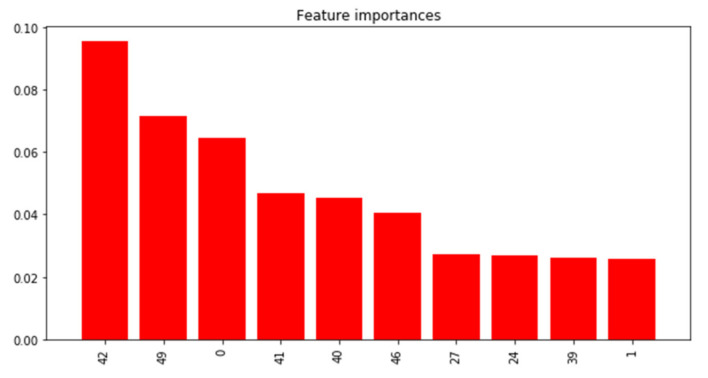
Feature importance for top-10 W-TSS shapelets predicting days with and without inhaler use in PRISMS.

**Figure 15 sensors-21-05801-f015:**
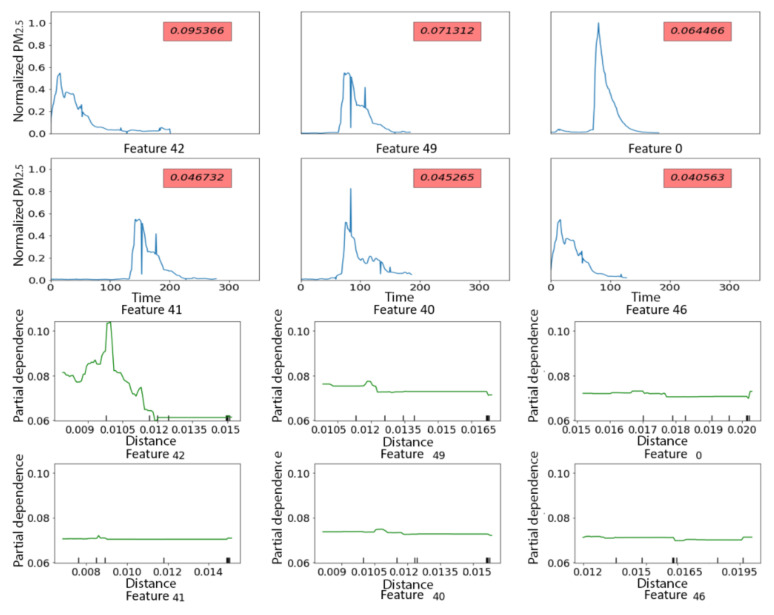
Top 6 shapelets predicting days with vs. without rescue medication in PRISMS based on residential indoor PM_2.5_ time series. Y-axis of partial independence plots: predicted probability. Red label in shapelet plots: feature importance value. Top panels display the shapelets along with their feature importance and bottom panels display partial dependence plots with x-axes representing the minimum distance to the shapelet and y-axies representing predicted probability of rescue medication use (vs. no use).

**Table 1 sensors-21-05801-t001:** Confusion matrix for predicting the four types of TRACE time series, with F_1_ score.

Test N = 100	True Class 1	True Class 2	True Class 3	True Class 4	Total
Predicted class 1	24	0	0	0	24
Predicted class 2	0	29	0	0	29
Predicted class 3	0	0	28	0	28
Predicted class 4	0	0	0	19	19
Total	24	29	28	19	F_1_ = 1

**Table 2 sensors-21-05801-t002:** Confusion matrix for predicting indoor PM_2.5_ vs. outdoor PM_2.5_, with F_1_ score.

Total Days = 544	True Indoor PM_2.5_	True Outdoor PM_2.5_	Total
Predicted Indoor PM_2.5_	266	0	266
Predicted Outdoor PM_2.5_	0	278	278
Total	266	278	F_1_ = 1

**Table 3 sensors-21-05801-t003:** Confusion matrix for predicting daily indoor PM_2.5_ time series with vs. without rescue medicine use, with F_1_ score.

Total Days = 272	True Use Days	True no Use Days	Total
Predicted use days	4	15	19
Predicted no-use days	8	245	253
Total	12	260	F_1_ = 0.26

## Data Availability

The datasets generated during and/or analyzed during the current study are available from the corresponding author on reasonable request. The data are not publicly available due to IRB concerns.

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
