# Peer review of "W-TSS: A Wavelet-Based Algorithm for Discovering Time Series Shapelets"

_sensors, 2021, doi:10.3390/s21175801_

Round 1
Reviewer 1 Report
The proposed new method is interesting and soundful, There are some problems with the organization and description of the paper, here i just listed some of them:
1.the title is not clear,the specific research object is not clear
2.'e.g., dynamic time warping methods) [1,2].’:chamge to [1-2], maybe need add more rfs
3.some english writting need improved, e.g.,TSS algorithms have the following basic steps: (1)
4.some figures are without details discussion, e,g, give more details on figure 2,
5. in section 3.1, how 32 and 64 be determined
6.Figure 4, 5 are hard to read, give sub-labels on the figure, maybe in colorful plots, also for figure 7
7.Figure 6:say something about learning TSS ?
Reviewer 2 Report
Reviewer’s Report on the manuscript entitled:
Wavelet-based discovery for time series shapelets
The authors propose a wavelet time series shapelet method for identifying candidate shapelets using wavelet transforms. They show the advantages of their method over some of the previous time-series shapelet methods via synthetic examples and series derived from residential air pollution sensors. They show W-TSS is more computationally efficient, more accurate, and can discriminative shapelets without the need for pre-specification of shapelet length. In my view, the paper is well-written and interesting and can be accepted after the following comments are addressed.
The quality of Figures must be improved:
The x-axis and y-axis labels are missing in almost all the Figures. Please insert the labels. Furthermore, the size of the labels and numbers must be enlarged and consistent to be more readable.
Line 135. Please add the following review paper on wavelets and time series analysis: https://doi.org/10.3390/app11136141 Please also add this reference also in line 153 for further detail and applications.
Line 247. The arbitrary selection could be an issue here. I suggest using some statistical methods, such as the elbow method or others for determining an optimal number of clusters, see this article for more details https://doi.org/10.1006/nimg.1998.0391
No need for change here just discuss this in the Discussion
Line 294. The flat shapelets may also represent the time series pattern and so removing all of them may bias the results. I think the training data set also requires some of the flat shapelets too?
Equation (1): it should be dt in the integral.
Lines 378-384 and lines 414 to 422 could go to a new section “Conclusion”. Also, please discuss the k-means and clusters in the Discussion along with the limitations of the study.
Please show a flowchart of the W-TSS in Section 2.2.
Please add an acronym table at the end of the manuscript.
Thank you for your contribution
Regards,
Round 2
Reviewer 1 Report
The author has take all my concerns into consideration and greately improved the manuscript. I have no other comments.
Author Response
Dear reviewer,
We would like to thank you for your great help in reviewing our paper. Your great efforts and valuable comments have really motivated us to make this manuscript into a much better shape.
Best regards!
Reviewer 2 Report
I would like to thank the authors for addressing my comments. The manuscript looks better now. Please see a few more comments below.
Optional comment but strongly encouraged: The label size of the figures is now better and more readable; however, the position of the label names (e.g., Time, Distance, Signal Value) could be moved to the centers instead of corners. In other words, the x-axis label (e.g., Time) usually goes in the middle like your previous Figure 3 not far to the right. Same as for the y-axis label. The authors may use paint.net or photoshop to manually correct the label positions!
Mandatory. The x-axis values in Figure 14 are overlapping and unreadable. I suggest removing some of those numbers, so the rest do not overlap.
I suggest authors revisit the figures to ensure that all the names and numbers are readable, consistent in size and format, and positioned correctly with a resolution of at least 300 dpi.
Line 448. Please replace the phrase “we also noticed that there are a few other” with “there are”, so please remove “we also noticed that” and “a few”.
Line 450. Please also add the phrase “Least-Squares Wavelet Analysis” right after “Constrained Least-Squares Spectral Analysis”.
Page 23. The flowchart (Figure A1) must be inserted in the body of the manuscript, not as an appendix. Furthermore, in one of the boxes of the flowchart it says “Powe Values”, but it should be “Power Values”.
Please carefully proofread the article before publication.
Thank you for your contribution
Regards,
